# Towards Enhancing Coding Productivity for GPU Programming Using Static Graphs

Leonel Toledo [1,*,†] , Pedro Valero-Lara [2,*,†] , Jeffrey S. Vetter [2,*] and Antonio J. Peña [1,*]

1    Barcelona Supercomputing Center (BSC)—Fundaciò i2CAT, 08034 Barcelona, Spain
2    Oak Ridge National Laboratory, Oak Ridge, TN 37831, USA
*    Correspondence: leonel.toledo@i2cat.net (L.T.); valerolarap@ornl.gov (P.V.-L.); vetter@ornl.gov (J.S.V.);
     antonio.pena@bsc.es (A.J.P.)
†    These authors contributed equally to this work.

**Abstract:** The main contribution of this work is to increase the coding productivity of GPU programming by using the concept of Static Graphs. GPU capabilities have been increasing significantly in terms of performance and memory capacity. However, there are still some problems in terms of scalability and limitations to the amount of work that a GPU can perform at a time. To minimize the overhead associated with the launch of GPU kernels, as well as to maximize the use of GPU capacity, we have combined the new CUDA Graph API with the CUDA programming model (including CUDA math libraries) and the OpenACC programming model. We use as test cases two different, well-known and widely used problems in HPC and AI: the Conjugate Gradient method and the Particle Swarm Optimization. In the first test case (Conjugate Gradient) we focus on the integration of Static Graphs with CUDA. In this case, we are able to significantly outperform the NVIDIA reference code, reaching an acceleration of up to 11× thanks to a better implementation, which can benefit from the new CUDA Graph capabilities. In the second test case (Particle Swarm Optimization), we complement the OpenACC functionality with the use of CUDA Graph, achieving again accelerations of up to one order of magnitude, with average speedups ranging from 2× to 4×, and performance very close to a reference and optimized CUDA code. Our main target is to achieve a higher coding productivity model for GPU programming by using Static Graphs, which provides, in a very transparent way, a better exploitation of the GPU capacity. The combination of using Static Graphs with two of the current most important GPU programming models (CUDA and OpenACC) is able to reduce considerably the execution time w.r.t. the use of CUDA and OpenACC only, achieving accelerations of up to more than one order of magnitude. Finally, we propose an interface to incorporate the concept of Static Graphs into the OpenACC Specifications.

**Keywords:** coding productivity; tasking; data dependencies; static graph; CUDA; OpenACC; conjugate gradient; particle swarm optimization

## 1. Introduction

It is undeniable that GPU capabilities have been increasing significantly in terms of performance and memory capacity. However, some applications are facing problems in terms of scalability and some algorithms seem to limit the amount of work that one GPU can perform at a given time [1]. This is mainly due to the assignment of hardware resources and the occupancy of the device, which makes it difficult to benefit from the whole GPU capacity. On the other hand, tasking allows algorithms with run-time dependent execution flows to be parallelized. Tasking provides a solution by implementing a queuing system, which manages all of the assignment of threads/kernels dynamically and decides which chunk of the work needs to be performed [2]. NVIDIA developed the CUDA Graph API as a potential solution to improve scalability. In CUDA Graph API, it is possible to represent the workflow as a graph, as an alternative for submitting kernels. These graphs are built

from a series of operations that could range from kernel invocations to memory copies, as well as host code or calls to libraries, such as CUBLAS and CUSPARSE. Every call inside the graph is represented as a node, and each node is connected by dependencies. This article is an extension of a conference publication focused on OpenACC [3]. In this work, we have included additional research using CUDA and a new application, the Conjugate Gradient method. The main contributions of this work are:

1.　Increase coding productivity of GPU programming by using Static Graphs, minimizing the overhead associated with the launch of kernels and maximizing the use of GPU capacity;
2.　Accelerations of up to more than one order of magnitude in OpenACC and CUDA applications;
3.　A new easy-to-use proposal for integration into the OpenACC Standard, which defines the use of Static Graphs into this programming model.

Those HPC applications that are composed of multiple memory bound kernels which have to perform the operations repeatedly or have an iterative nature, such as those leveraged in this work, but many others as well, such as CFD simulations [4–7], image processing [8,9], AI kernels [10], or Linear Algebra kernels [11–14], just to mention of few, can benefit from the use of Static Graphs by reducing the CPU–GPU communication overhead and achieving higher GPU occupancy. To the best of our knowledge, this is the first time that CUDA Graph has been integrated with OpenACC and effectively adapted to the two different algorithms used as test cases in this work: the Conjugate Gradient Method and Particle Swarm Optimization.

The rest of this document is organized as follows: Section 2 describes: (i) the programming models used in this paper, CUDA and OpenACC and (ii) the most important concepts about the new CUDA API to implement Static Graphs. Section 3 presents a detailed analysis of the implementations and performance achieved by using CUDA Graph in combination with CUDA and OpenACC on the two algorithms studied, Conjugate Gradient (CUDA) and Particle Swarm Optimization (OpenACC). In Section 5, we propose a new specification to use Static Graphs in the OpenACC Standard. Section 6 presents the most relevant state-of-the-art references. Section 7 concludes with the most important remarks and proposes future directions.

## 2. Background

### 2.1. CUDA

CUDA (Compute Unified Device Architecture) is a parallel computing API developed by NVIDIA, which allows developers to use GPUs for general purpose processing and provides support for languages such as C, C++ and Fortran. CUDA allows developers to write programs targeted for GPU, using functions that resemble C code. However, there are important differences that must be addressed. For instance, a function (kernel) that is compiled for GPUs is executed in one or a set of streaming multiprocessors (SM), depending on the number of CUDA threads necessary for the computation. One of the key features of the hardware is that cores inside a SM follow the Single Instruction Multiple Data (SIMD) principle, where groups of threads execute the same instruction concurrently, but with different data.

Historically, GPUs are not very well-suited to parallelizing the computation of several independent kernels in the same GPU [15,16] to efficiently exploit the bigger and bigger GPU capacity. One example which attempted to maximize the use of the GPU resources was CUDA Dynamic Parallelism. Using Dynamic Parallelism the programmers could invoke kernels inside the device without the need for switching context back to the CPU. However, the launching of kernels from other kernels has a large associated computational cost [1].

### 2.2. OpenACC

OpenACC is a high-level, directive-based programming model which supports C, C++ and Fortran. It was developed to allow programmers to interact with heterogeneous

HPC architectures without the effort that requires to fully understand all the low-level programming details and underlying hardware features [17]. This programming model allows developers to insert hints into their code that help the compiler to interpret how to parallelize the code. In this way, the compiler is responsible for the transformation of the code to make it parallel, which is completely transparent to the programmer.

OpenACC defines a mechanism to offload programs to an accelerator in a heterogeneous system [18]. Because OpenACC is a directive-based programming model, the code can be compiled serially, ignoring the directives and still produce correct results, allowing a single code to be portable across different platforms [19]. This simple model allows non-expert programmers to easily develop code that benefits from accelerators [20]. Currently, OpenACC compilers support several platforms such as ×86 multicore platforms, accelerators (GPUs, FPGAs), OpenPOWER processors, KNL and ARM processors. One example that summarizes the advantages of using OpenACC is the the work of [21], which evaluates the use of OpenACC, OpenCL and CUDA in terms of performance, productivity, and portability. This work concludes that OpenACC is a robust programming model for accelerators while improving programmer productivity.

*2.3. CUDA Graph API*

The performance of GPU architectures continues to increase each generation. However, it is important to address that each kernel launch has an associated overhead regarding the submission of each operation to the GPU. These overheads are becoming more and more significant and can have a negative impact on performance [22]. Many current applications need to perform a large number of different operations to solve a given problem. Most of the times these operations are involved in patterns that require many iterations, so this kind of overhead can produce a significant performance degradation.

To address this issue, since CUDA 10.0, it is possible to represent the workflow as a graph. A graph consists of a series of operations such as memory copies and kernel launches, which are connected by dependencies. This feature allows developers to represent the work as a graph of nodes and create a static structure that can be launched at any time and be executed as many times as needed. CUDA Graph API has two main advantages: First, the overhead of launching GPU operations, such as memory transfers or kernel executions, has no impact on performance, since the static structure which defines the graph is submitted only once to the GPU. Second, we have the freedom to create the workflow to be submitted to the GPU. There may be operations which are completely independent from each other, so depending on the hardware capabilities, it is possible to overlap the execution of different nodes of the graph.

## 3. Use Case I: Conjugate Gradient

The Conjugate Gradient (CG) is a well-known and widely used iterative method for solving sparse systems of linear equations. These systems arise in many important settings, such as finite difference and finite element methods, partial differential equations, structural analysis, circuit analysis, and many more linear algebra related problems [23]. These kinds of problems, due to the particular characteristics of the CG method, can benefit from using the CUDA Graph API. Some of the most time consuming steps of this method may efficiently be overlapped, reducing considerably the execution time and the overhead associated to the launch of multiple kernels. CG is mainly composed of the next major kernels or operations (note that all operations are performed in double precision):

- The dot product (cublasDdot in Listings 1 and 2) is the sum of the products of the corresponding components of the vectors of the same size;
- The AXPY vector-scalar product (cublasDaxpy in Listings 1 and 2) is a combination of scalar multiplication and vector addition. This computes $y + \alpha x$, where $y$ and $x$ are vectors and $\alpha$ is a scalar;
- The sparse matrix–vector multiplication (cusparseDcrsmv in Listing 1 and cublasD-spmv in Listing 2) consists in computing a matrix–vector multiplication where the

matrix is sparse (the number of zeros in the coefficients of the matrix is grater than the non-zeros coefficients);

- Other major steps comprise very simple operations such as the division of two scalars (r1_div_x) or the copy of the components of one vector to another vector (cudaMemcpy).

**Listing 1.** NVIDIA reference CG code using CUDA Graph [24]. Note that in the original code the operations are performed using single precision, but in our analysis we used double precision.

```
cudaGraph_t graph;
cudaGraphExec_t instance;
cudaStream_t stream1;

cudaStreamBeginCapture(stream1, cudaStreamCaptureModelGlobal);
d_b = r1_div_x<<<..., stream1>>>(d_r1, d_r0);
cublasDscal(d_b, d_p)
cublasDaxpy(cublasHandle, alpha, d_r, d_p);
cusparseDcsrmv(cusparseHandle, A, vecp, vecAx,);
memset(d_dot, 0.0);
d_dot = cublasDdot(d_p, d_Ax);
d_a = r1_div_x<<<..., stream1>>>(d_r1, d_dot);
cublasDaxpy(cublasHandle, d_a, d_p, d_x);
a_minus<<<..., stream1>>>(d_a, d_na); //d_na = d_a − 1
cublasDaxpy(cublasHandle, d_na, d_Ax, d_r);
cudaMemcpyAsync(d_r0, d_r1, DeviceToDevice, stream1);
cudaMemsetAsync(d_r1, 0.0, stream1);
d_r1 = cublasDdot(cublasHandle, d_r, d_r);
cudaMemcpyAsync(condition, d_r1, DeviceToHost, stream1);
cudaStreamSynchronize(stream1);
cudaStreamEndCapture(stream1, graph);
cudaGraphInstantiate(instance, graph);

while (condition > tolerance^2  && k <= max_iter)
{
  cudaGraphLaunch(instance, stream);
}
```

*3.1. NVIDIA Conjugate Gradient*

Leveraging the CUDA Graph API, we present an optimized version of the CG using as a base code for this study the reference NVIDIA code for CG [24]. The different major steps of this code are shown in Listing 1. In this code, each operation is executed sequentially and each kernel must wait until the previous is finished, to start executing.

When using the CUDA Graph API to convert the baseline code as shown in Listing 1, we obtain as a result a graph with 14 nodes, which are computed in each iteration. The only potential gain at this point is reducing the overhead related to kernels launching. As shown in Figure 1-left, some of these operations feature dependencies from the previous kernels and must wait until the data is ready to be consumed. Those operations that may be computed in parallel correspond to memory transfers and kernels. The time associated to these tasks is very different, which makes it difficult to obtain any benefit from the use of CUDA Graph.

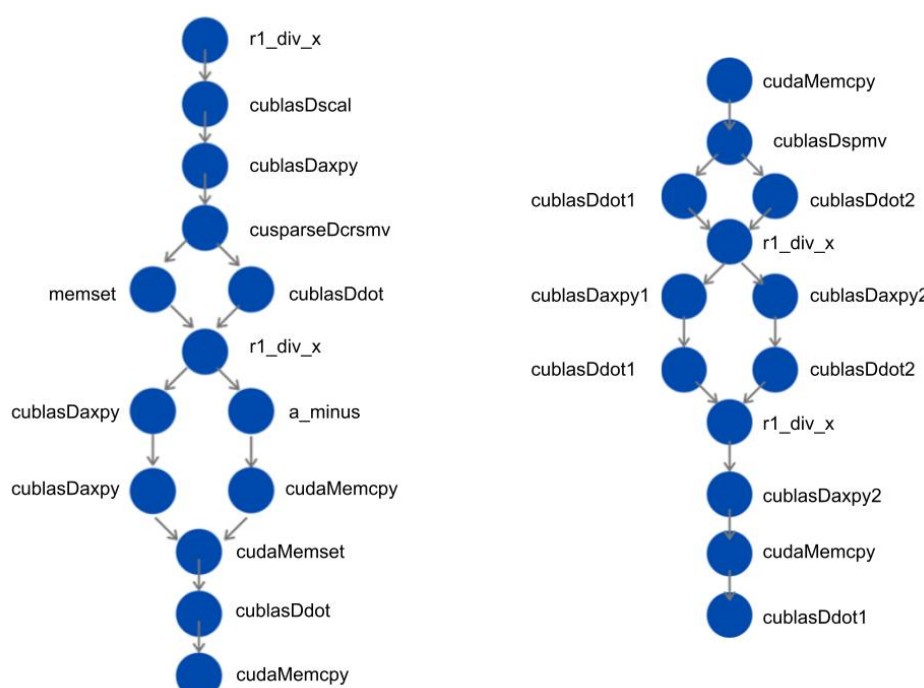

**Figure 1.** Graph configuration for the NVIDIA CG reference code (**left**) and optimized approach (**right**).

### 3.2. Optimized Conjugate Gradient Method

We reformulated the algorithm and its corresponding representation using CUDA Graph to benefit from those operations which can be overlapped and executed in parallel. In that way, we are configuring the algorithm to obtain the most potential speedup and kernel overlapping. Listing 2 shows the modifications in the algorithm. These modifications consist of swapping the order of execution of some of the kernels, mainly AXPY operations and DOT products [12], exposing a higher parallelism.

Figure 1-right shows how the new structure of the graph looks when using a more appropriate algorithm. This new configuration has the potential for higher execution efficiency due to the work being distributed in a way where the most computationally expensive operations may perform in parallel.

### 3.3. Performance Analysis

We conduct the performance evaluation by using the following heterogeneous system: 2 × IBM Power9 8335-GTH at 2.4 GHz, 32 GB RAM memory, and an NVIDIA V100 (Volta) GPU with 16 GB HBM2 and NVLink2 for high-bandwidth communication between CPU and GPU. This architecture is similar to that used in the current top-2 (Summit at ORNL) and top-3 (Sierra at LLNL) fastest supercomputers in the TOP500 list. We analyzed the major steps of the NVIDIA reference code (Listing 1) and determined that about 54% of the execution time is spent in the computation of the major operations/steps of the algorithm. Those major operations that may be run in parallel represent most of the execution time, which provides a potential improvement. In our tests, we have used several configurations of randomly generated diagonal dominant tri-diagonal matrices of different sizes. This is a very common setting used for the performance analysis of this kind of operation [25]. We identified that both AXPY and DOT products, as well as the SpMV operation, are the bottlenecks and the most computationally expensive calculations in the algorithm. Unlike SpMV, the operations AXPY and DOT product can be computed in parallel. The number of iterations that was required until the solution was converged was the same in both codes regardless the size of the matrices, with just a couple of exceptions where the optimized code took at most one more iteration to converge. On relatively small

size matrices (matrix size < 1,000,000 × 1,000,000), it took three iterations until solution convergence was achieved. For matrix sizes larger than one million, it took between eight and ten iterations to converge. It is important to note that the matrices used in the experiments are the main diagonal (well-conditioned) matrices. That is why a low number of iterations was necessary to converge; however, one can expect a higher number of iterations on another type of matrices. Depending on the problem, especially for those that require a large number of iterations to converge, the impact of the proposed optimizations is even more significant. The time to create and instantiate the graph is relatively large, around 400–500 µs, but this is only performed once, at the beginning of the execution, being a very small cost compared to the time consumed by the kernels.

Figure 2-left shows the average time in seconds of computing one single iteration of the algorithm, comparing the NVIDIA reference code against the proposed code that implements a better kernels distribution and potential overlapping. The blue bar represents the results obtained by the optimized algorithm using CUDA Graph, the red bar represents the execution time of the NVIDIA reference code. Finally, in order to see the impact of using CUDA Graph to potentially overlap the computation of different kernels and reduce the overhead associated to the launch of these, in the yellow bar we show the execution time of the optimized algorithm without using CUDA Graph. It is important to mention that regardless the case, the optimized version using CUDA Graph was always the fastest, performing an acceleration ranging from 10% to 30% with respect to the optimized version without CUDA Graph. The use of CUDA Graph is not only an important increase in performance but also an increase in programming productivity. This additional performance is reached by *only* adding a few lines of code corresponding to the creation of the graph and declaration of the nodes.

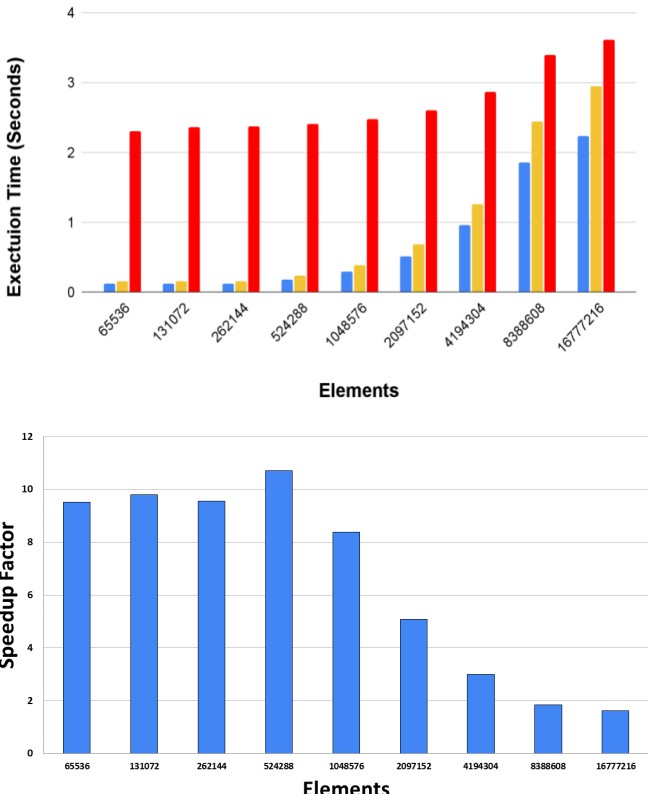

**Figure 2.** Top figure: execution time of the three different CG versions: NVIDIA reference code (Listing 1) in red color, optimized version using CUDA Graph (Listing 2) in blue color, and optimized version without using CUDA Graph (in yellow color). Bottom figure: speedup achieved by the optimized version using CUDA Graph (Listing 2) w.r.t. the NVIDIA version (Listing 1).

**Listing 2.** Optimized CG code using CUDA Graph.

```
// Solve Ax=b using Conjugate Gradient method
cudaGraph_t graph;
cudaGraphExec_t instance;
cudaStream_t stream1, stream2;
cudaEvent_t kernelEvent1, kernelEvent2;
// Initial setting
cudaMemcpy(r, b, DeviceToDevice);
cudaMemcpy(p, r, DeviceToDevice);
//stream1 is the origin stream
cudaStreamBeginCapture(stream1, cudaStreamCaptureModelGlobal);
cudaMemcpyAsync(rOld, r, DeviceToDevice,stream1);
cublasDspmv(cublasHandle, A, p, s);
alpha1 = cublasDdot1(r, r);
alpha2 = cublasDdot2(p, s);
alpha = r1_div_x<<<..., stream1>>>(alpha1, alpha2);
//AXPY Fork
cublasDaxpy1(cublasHandle, alpha, p, x, stream1);
cudaEventRecord(kernelEvent1, stream1);
cudaStreamWaitEvent(stream2, kernelEvent1);
cublasDaxpy2(cublasHandle, -alpha, s, r, stream2);
//Join stream2 back to stream1
cudaEventRecord(kernelEvent2, stream2);
cudaStreamWaitEvent(stream1, kernelEvent2);
//DOT product Fork
beta1 = cublasDdot1(cublasHandle, r, r, stream1);
cudaEventRecord(kernelEvent1, stream1);
cudaStreamWaitEvent(stream2, kernelEvent1);
beta2 = cublasDdot2(cublasHandle, rOld, rOld, stream2);
//Join stream2 back to stream1
cudaEventRecord(kernelEvent2, stream2);
cudaStreamWaitEvent(stream1, kernelEvent2);
beta = r1_div_x<<<..., stream1>>>(beta1, beta2);
cudaMemcpyAsync(rAux, r, DeviceToDevice, stream1);
cublasDaxpy2(beta, p, rAux);
cudaMemcpyAsync(p, rAux, DeviceToDevice, stream);
condition = cublasDdot(r, r);
cudaStreamSynchronize(stream1);
cudaStreamEndCapture(stream1, graph);
cudaGraphInstantiate(instance, graph);

while (condition > tolerance^2 && k <= max_iter)
{
  cudaGraphLaunch(instance, stream1);
}
```

Figure 2-right graphically illustrates the speedup achieved by the optimized version using CUDA Graph with respect to the NVIDIA reference code. The optimized code is able to achieve an acceleration higher than $10\times$ with respect to he NVIDIA reference code in some cases. The size of the matrices/vectors has an important impact in performance. On the largest sizes computed, we see a lower speedup (about $2$–$5\times$). This is mainly because of (i) the potential overlapping of kernels (AXPY and DOT product) is more difficult on larger vectors, since the computations to be done require more computational resources and (ii) the impact of the time consumed by the launch of kernels is less important when running more time-consuming kernels. However, on the other sizes, a very high benefit is shown (close to $11\times$ faster). This is possible because of a better distribution of the components of the algorithms, which allows to expose a higher parallelism and can be better mapped on

GPU resources. Unfortunately, current profiling tools, such as the NVIDIA Visual Profiler or Nsight, do not provide information on kernel overlapping when using CUDA Graph, so we are unable to visually show this effect.

## 4. Use Case II: Particle Swarm Optimization

Particle Swarm Optimization (PSO) is an evolutionary computational technique originally developed by Kennedy and Eberhart [26]. The algorithm was developed as a simulation of a simplified social system, with the objective to simulate the behavior of bird flocks. This algorithm is also considered as an optimizer. This technique shares some similarities with genetic algorithms. For instance, the system is initialized with a random population that evaluates different sets of solutions. Every potential solution is considered as a particle within the search space of the problem. This means that each particle has its own set of parameters such as velocity, speed, acceleration, position and learning factors. Each particle keeps track of the values which are associated with the best solution, also known as fitness. This is an iterative algorithm, where in every iteration, the values (speed, position, etc.) of all the particles are evaluated and changed. These values are changed with the target of moving each particle to locations that potentially have a better solution. PSO is used by many applications; for instance, those problems that involve maximization or minimization [27,28]. PSO is robust enough to work with functions in a continuous, discrete or mixed search space, as well as multi-objective problems [29].

In this work we use PSO as a case of study to test the impact of integrating static graphs on directive-based programming models. Due to the iterative design of the algorithm and its potential to parallelize several areas of the code, it is an extraordinary test bed for studying and analyzing the impact of the use of GPU static graphs. It is important to mention that the target of this work is not to improve the PSO algorithm itself, but to study the impact of combining the different target programming models (OpenACC and CUDA Graph) in an application. PSO is a stochastic problem that is used to optimize mathematical functions within a delimited search space. This technique is widely used for machine learning, being similar to other annealing and genetic algorithms. For the scope of this research, we define particles in a 3D space; each particle represents a possible solution of a target function. Each particle moves within the search space and evaluates a function until a number of iterations is achieved or the particles converge at the optimum value. In the context of this study, we analyze the performance of the algorithm and programming models using different values for the size of the population, the objective function, the type of optimization (maximum and minimum) and the maximum allowed iterations. To measure the impact of the different approaches, we developed several versions of PSO. First, we studied a sequential version of the algorithm, which we use as a baseline. The code that we tested is shown in Listing 3.

We implemented the original version of the code based on the work of Kennedy et al. [27]. In the first step we initialize all particles by defining their position and velocities within the boundaries of the search space. Then, each particle evaluates the solution determined by its position to calculate its fitness. Next, we find the best fitness of the population and store its value. The following steps are computed in each iteration: (i) calculate the position of the particle that has the best fitness; (ii) compute the next position; and (iii) update each particle speed in the x, y and z axes. Afterwards, we calculate the fitness of the particles in their new position and finally we update the value of the best particle. We repeat these steps as many iterations as needed, and finally the particle with the best fitness is selected as the best possible solution of the problem. Given the stochastic nature of the problem, it is not guaranteed to find the optimum value all the times. However, sometimes an approximation is enough and it has the advantage of being faster than a brute force search.

**Listing 3.** OpenACC PSO code.

```
void main () {
  // Initialization
  initParticle(array_population);
  calculateFitness(array_population);
  updatePopulationBest(array_population);
  // Computation
  while(i<ITERATIONS) {
    // findBestParticle kernel
    #pragma acc kernels deviceptr(array_population)
    for(int i=0; i<POPULATION; i++)
      findBestParticle(array_population[i]);
    // updateParticlePosition kernel
    #pragma acc kernels deviceptr(array_population)
    for(int i=0; i<POPULATION; i++)
      updateParticlePosition(array_population[i]);
    // calculateFitness kernel
    #pragma acc kernels deviceptr(array_population)
    for(int i=0; i<POPULATION; i++)
      calculateFitness(array_population[i]);
    // updateBestPopulation kernel
    #pragma acc kernels deviceptr(array_population)
    for(int i=0; i<POPULATION; i++)
      updateBestPopulation(array_population[i]);
  }
}
```

*4.1. OpenACC and CUDA Graph Implementations of PSO*

The use of GPUs has many advantages in terms of acceleration and performance compared to the sequential version of the code. However, as mentioned previously, there are some problems that prevent the algorithm from benefiting from the whole hardware/ architecture capacity. To achieve additional acceleration, we involved the GPU for computing most of the major tasks in the algorithm, as well as avoiding having many small tasks that are constantly switching context and awaiting for some of them to finish. We use OpenACC for GPU parallelization (see Listing 3).

The modifications of the code from the CPU to the GPU architecture are minimum. However, there are important considerations that significantly impact the behavior of the code. The main difference is in the distribution of the work. Another significant factor is the use of Unified Memory. Managing memory between CPU and GPU is an important challenge. There are significant limitations, particularly concerning memory bandwidth, latency and GPU utilization [30]. To mitigate this issue, since CUDA 6.0 it has been possible to use Unified Memory access. This provides a mechanism to simplify the GPU memory communication with the host while providing high bandwidth for data transfers at runtime. We also use Unified Memory for more readable code and coding productivity [31]. Doing this, the GPU and CPU memory communications can be kept hidden from the developer, so the programmer does not have to deal with the issues that arise from moving data, further enhancing coding productivity.

The OpenACC version can be efficiently integrated with CUDA Graph to minimize the overhead of creating and launching multiple kernels in every iteration. Although this overhead is measured at the scale of microseconds, this can degrade the performance considerably on long runs. Using CUDA Graph we can create a high-level representation of the workflow; in other words, we determine the topology of the graph by determining the order of the tasks that need to be executed in every iteration. We still use the OpenACC kernels as a node of the graph. The code presented in Listing 4 shows the changes that were made in order to combine both models—OpenACC and CUDA Graph.

**Listing 4.** OpenACC and CUDA Graph PSO code.

```c
int main (int argc, char *argv[])
{
  cudaGraph_t graph;
  cudaGraphExec_t instance;
  cudaStream_t stream1, stream2;
  cudaEvent_t event1, event2;
  // Initialization
  initParticle(array_population);
  calculateFitness(array_population);
  updatePopulationBest(array_population);
  // Graph definition
  cudaStreamCreate(&stream1);
  cudaStreamCreate(&stream2);
  void* stream = acc_get_cuda_stream(acc_async_sync);
  acc_set_cuda_stream(0, stream1);
  cudaStreamBeginCapture(stream1, cudaStreamCaptureModeGlobal);
  // OpenACC Kernels
  findBestParticle(array_population, stream1);
  // Fork
  cudaEventRecord(event1, stream1);
  updateParticlePosition(array_population, stream1);
  calculateFitness(array_population, stream2);
  // Join
  cudaEventRecord(event2, stream2);
  cudaStreamWaitEvent(stream1, event2);
  updateBestPopulation(array_population, stream1);
  cudaStreamEndCapture(stream1 , &graph);
  cudaGraphExec_t graphExec;
  cudaGraphInstantiate(&graphExec, graph, NULL, NULL, 0);
  //Computation
  for (int i = 0; i < ITERATIONS; i++)
  {
    cudaGraphLaunch(graphExec, stream1);
  }
}
```

CUDA Graph allows us to store the set of kernels to be computed (workflow) before being launched. In that way, it is possible to know the amount of work that needs to be submitted to the GPU in advance. To achieve that, we use the OpenACC acc_get_cuda_stream(acc_async_sync) and acc_set_cuda_stream(0,stream1) instructions. In that way we ensure that CUDA Graph recognizes the streams used by OpenACC. Finally, this stream must also be known by the OpenACC async clause. Using OpenACC and CUDA Graph, the stream creations are more efficient and execution is faster. This is due to the way that CUDA Graph launches the kernels to the GPU. All the kernels are treated as a whole instead of processing each of them individually. This considerably reduces the overhead when submitting multiple kernels to the GPU. All this allows us to achieve an important acceleration of up to 3x with respect to the pure OpenACC implementation, at the expense of losing the coding productivity and high level of expressiveness of a pure OpenACC approach in this portion of the code. Finally, we also exploit the potential overlapping of those parts of the application which are independent and can be executed in parallel. These are the functions updateParticlePosition and calculateFitness. To do that—as we did in the Conjugate Gradient method (see Listing 2)—we need to use cudaEventRecord and cudaStreamWaitEvent.

### 4.2. Performance Analysis

Table 1 shows the details of the seven functions used in our analysis. These are considered as standard benchmarks for PSO. For the sake of simplicity, these formulas are represented for 1D space; however, all the functions used in our experiments were implemented for a 3D space.

**Table 1.** Description of the employed functions for testing PSO.

| Function | Formula |
|---|---|
| Sphere | $f(x) = \sum_{i=0}^{n} x_i^2$ |
| De Jong | $f(x) = \sum_{i=0}^{n} i * x_i^4$ |
| Griewank | $f(x) = \frac{1}{4000} \sum_{i=0}^{n} x^2 - \prod_{i=0}^{n} cos(\frac{x}{\sqrt{i}}) + 1$ |
| Rastrigin | $f(x) = \sum_{i=0}^{n} [x^2 - 10cos(2\pi * x_i) + 10]$ |
| Rosenbrock | $f(x) = \sum_{i=0}^{n} [100(x_{i+1} - x_i^2)^2 + (x_i - 1)^2]$ |
| Schaffer | $f(x) = \sum_{i=0}^{n-1} [0.5 + \frac{(sin\sqrt{x_i^2 + x_{i+1}^2}) - 0.5}{(1 + 0.001(x_i^2 + x_{i+1}^2))^2}]$ |
| Schaffer 2 | $f(x) = (\sum_{i=0}^{n} x_i^2)^{0.25} [1 + (50(\sum_{i=0}^{n} x^2)^{0.1})^2]$ |

For the experiments, we use all the functions described in Table 1 on a simulated population of 1000 particles. We execute all the simulations during 10,000 iterations. Figure 3 illustrates the wall time (μs) of our test bed. As expected, the sequential version is the slowest. We see a much higher performance by using the native OpenACC implementation. However, the hardware is used in a more efficient way with the combination of both programming models, OpenACC and CUDA Graph. This approach is able to achieve an important reduction in the execution time (even more than one order of magnitude in some cases). On average, the speedup is ranged from 2× to 4×.

It is important to highlight that our approach based on GPU static graphs (OpenACC and CUDA Graph) is very close to the optimized CUDA performance. The maximum performance difference between the optimized CUDA implementation and the OpenACC and the CUDA Graph counterpart is about 10%. As shown (Figure 3), the use of GPU static graphs (CUDA Graph) is specially well suited for iterative problems with fine-grained kernels, which have to be computed every iteration.

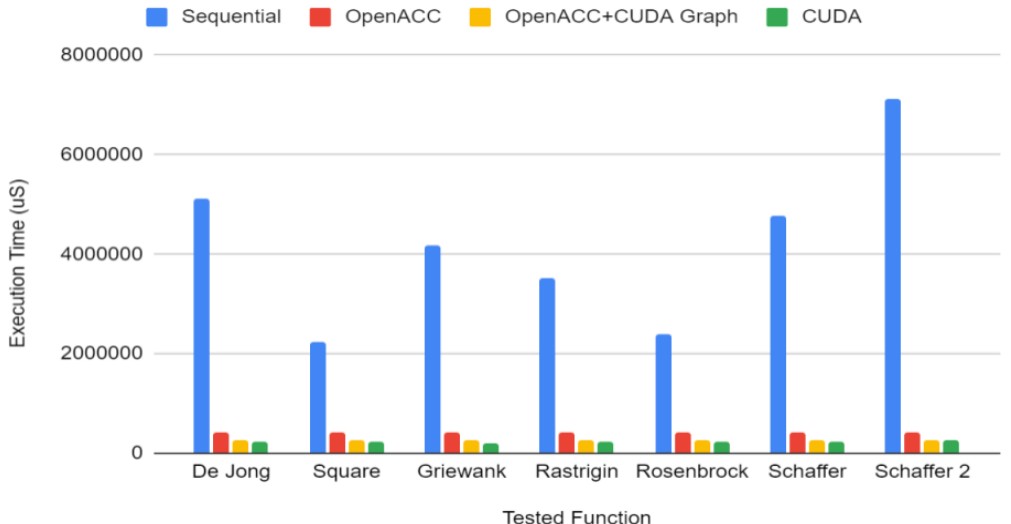

**Figure 3.** Execution time (μs) comparison between sequential, OpenACC, OpenACC and CUDA Graph and CUDA implementations of the PSO algorithm.

From this point on, we focus on the comparison between the OpenACC implementation and the use of GPU static graphs (CUDA Graph) as part of the OpenACC specification. The behavior illustrated in Figure 3 remains true in these new results, i.e., the sequential version continues being the slowest approach and the use of OpenACC and CUDA Graph is not farther away than 10% from the performance achieved by the optimized CUDA code. We make use of the Schaffer 2 function (Table 1) to test how the use of different settings can affect the behavior of our proposed model (GPU static graph). We decided to use this function because it is the most computationally expensive. First, we analyze the impact of increasing the size of the population (number of particles). In the PSO algorithm, the larger the population, the larger the kernels (more threads are necessary). Figure 4 illustrates the impact of increasing the number of particles on execution time. In these tests, the number of iterations is 100. The use of OpenACC and CUDA Graph is able to achieve an acceleration of up to $3.5\times$. However, the larger the population, the lower the acceleration. This is expected because the chance to execute more than one kernel in parallel in the GPU is reduced by increasing the number of particles. When running larger kernels, the acceleration reached is up to $1.3\times$.

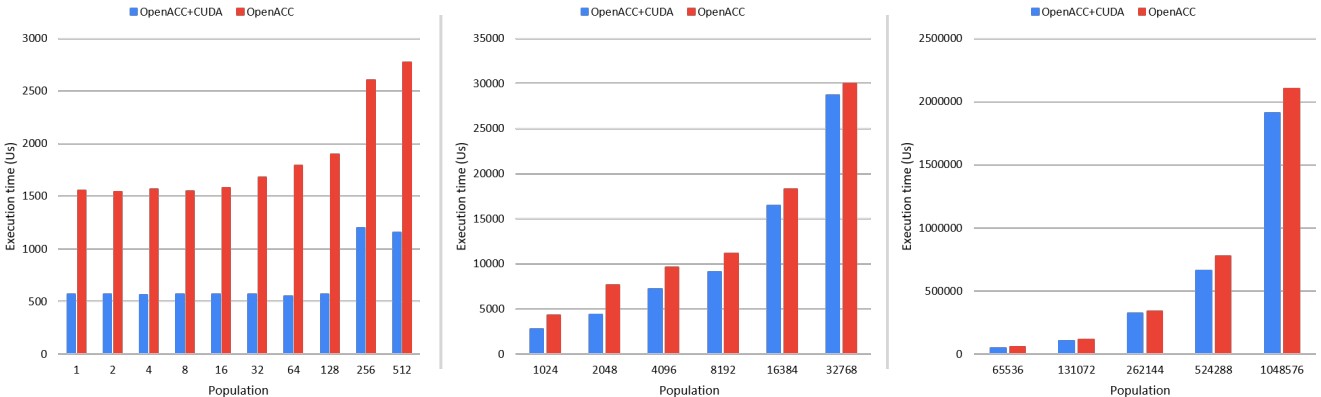

**Figure 4.** Execution time (µs) increasing the size of the population and keeping the number of iterations (100) constant.

Next, we analyze the impact on the performance of increasing the number of iterations. In PSO, the larger the number of iterations, the more kernels need to be executed to finish the simulation. In the experiments, we use a population equal to 1024 particles. The speedup reached in the previous experiments when using this size of population is $1.7\times$. Figure 5 illustrates the execution time for different test cases using a different number of iterations while keeping the size of the population constant. The use of CUDA Graph and OpenACC is able to keep the speedup in the range $1.4$–$1.7\times$ for a population size of 1024. A similar trend is reached when using different population sizes.

As we have seen along this section, it is possible to obtain an important acceleration by combining OpenACC with CUDA Graph, at the expense of some losing of coding productivity, which is against the motivation behind the OpenACC Standard. To mitigate this limitation, in the next section we present a proposal to integrate the concept of static graph into the OpenACC syntax.

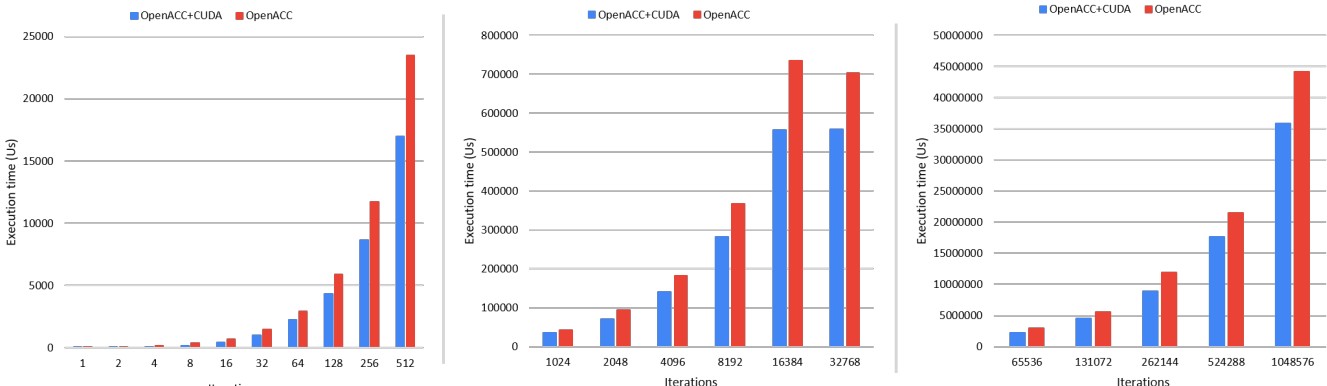

**Figure 5.** Execution time (μs) increasing the number of iterations and keeping constant the size of the population (1024 particles).

## 5. Directive-Based GPU Static Graph API Proposal

In the previous sections, we were able to prove the efficiency of using GPU static graphs (CUDA Graph) with CUDA and OpenACC. Although the performance was satisfactory, the integration of CUDA Graph with OpenACC is not easy, harnessing programming productivity. It is important to highlight that the time to develop parallel solutions is valuable factor to be considered along with portability. That is why it is so important to provide an efficient way to easily implement GPU codes while hiding low-level hardware/software details, which are usually very time-consuming. The CUDA programming framework requires developers to be familiar with the underlying architecture to obtain high efficiency on NVIDIA GPUs [32]. The implementation of CUDA codes is highly time-consuming, in particular when legacy codes need to be completely ported to CUDA. CUDA codes are also hard to maintain and port [33]. To deal with these constraints, OpenACC provides a simple directive-based model that aims to achieve a similar performance to using CUDA. Another important advantage of using OpenACC is that OpenACC codes are designed to be platform agnostic.

In this section, we propose a new approach for developers to use static graphs within OpenACC. Our motivation is to provide a simple and easy to use directive which can annotate and define a workflow as a static graph. We introduce a new pragma clause into the OpenACC programming model which can abstract developers from all the CUDA Graph related programming. Listing 5 shows an example of the code we propose using `static_graph pragma` for the PSO implementation.

The `static_graph` clause is interpreted by the compiler to create a static graph which treats every OpenACC kernel as a CUDA Graph node. The topology of the workflow is recorded using CUDA Graph and an instance of this can be run for as many iterations as necessary (`accGraph_t`). This poses an important reduction in the number of lines of code and a substantial increase in terms of programming productivity, in comparison with the original CUDA Graph and OpenACC code (see Listing 4). We use Unified Memory to further simplify the code, and avoid complex transfers of data between device and host. To exploit the potential overlapping among those parts of the application that can be executed in parallel, we need to use OpenACC Queues and the async clause, hence preventing the explicit handling of any CUDA constructs such as CUDA Streams. Figure 6 graphically illustrates a possible mapping of the OpenACC static graph API on the CUDA programming model.

**Listing 5.** OpenACC staticgraph model with kernels overlapping.

```
int main (int argc, char *argv[]) {
  accGraph_t graph;
  ...
  #pragma acc static_graph(graph) deviceptr(array_population)
  {
    // Enqueue
    #pragma acc kernels deviceptr(array_population) async(1)
    {
      findBestParticle(array_population);
    }
    // Fork & enqueue
    #pragma acc kernels deviceptr(array_population) async(1)
    {
      updateParticlePosition(array_population);
    }
    #pragma acc kernels deviceptr(array_population) async(2)
    {
      fitnessBestParticle(array_population);
    }
    // Join & enqueue
    #pragma acc kernels deviceptr(array_population) async(1)
    {
      updateBestPopulation(array_population);
    }
  } // End pragma acc static_graph
  for (int i = 0; i < ITERATIONS; i++)
    #pragma acc launch_static_graph(graph) deviceptr(array_population)
}
```

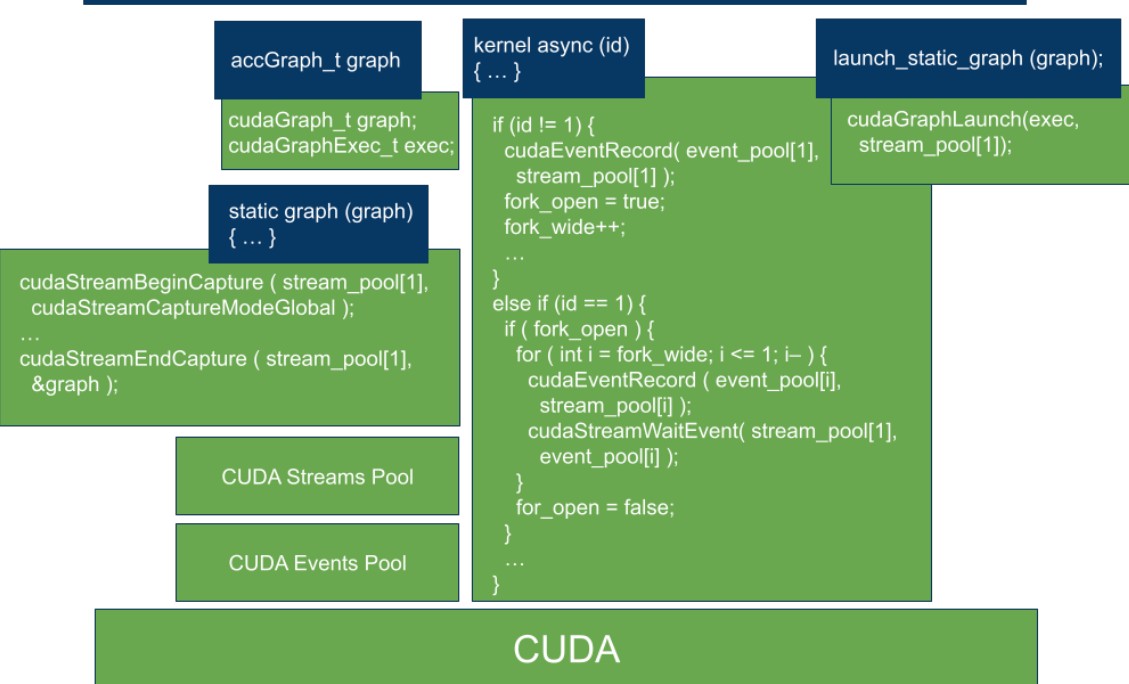

**Figure 6.** OpenACC static graph API mapping on the CUDA programming model.

These modifications proposed for the OpenACC specification provide developers with a robust and strong mechanism to easily translate iterative algorithms into static graphs. These graphs can be recorded prior to execution, which allows the runtime to be aware of the dependencies and the order of the execution. Once the topology is defined, then all the GPU work is handled as one single GPU launch by the driver, avoiding the overhead associated to deal with each of the kernels separately. As shown in previous sections, this yields significant benefits both in terms of performance and coding productivity (see Table 2).

**Table 2.** Overall performance benefit using Static Graphs.

| Application | |
| --- | --- |
| **Conjugate Gradient** | |
| NVIDIA Ref./CUDA + Static Graph | CUDA/CUDA + Static Graph |
| $2\times$–$11\times$ | $\approx 1.3\times$ |
| **Particle Swarm Optimization** | |
| OpenACC/OpenACC + Static Graph | CUDA/OpenACC + Static Graph |
| $1.3\times$–$4\times$ | $0.9\times$–$0.95\times$ |

## 6. Related Work

Although GPU capacity has increased significantly, the scalability of algorithms and applications still faces important challenges [1]. One important problem regarding scalability is the hardware resource assignment. Some applications are limited to execute a single kernel in the GPU without benefiting from the whole capability of the device [15,16,34]. The work of Pallipuram et al. [35] compares different programming models using several optimizations to evaluate performance of neural networks using different HPC and GPU architectures. The work of Memeti et al. [36] presents a detailed study of productivity, performance and energy consumption for different parallel programming models such as OpenMP, OpenCL, CUDA and OpenACC. It concludes that programming with OpenCL or CUDA requires more work and effort than programming with OpenACC. Also, the use of a fewer number of code lines often translates in less energy consumption. The work of Ashraf et al. [37] explores the combination of homogeneous and heterogeneous frameworks into energy efficient GPU devices for exascale computing systems. This is by comparing quantified metrics of CUDA subroutines against KAUST linear algebra subprograms. Other interesting examples of related works are the task-based programming models using GPUs, such as StarPU [38] and OmpSs [39]. Both programming models propose a task-based API which allows tasks to be executed on GPUs and tune scheduling algorithms. The work of Kato et al. [40] proposes a GPU scheduler to provide prioritization and isolation capabilities for GPU applications in real-time and multi-tasking environments. In contrast, the focus of our work is twofold: (i) analyze the potential of using CUDA Graph for an enhanced programming productivity, performance and scalability of applications; and (ii) propose a new specification for a better integration of OpenACC with CUDA Graph.

## 7. Conclusions and Future Work

In this work, we present how CUDA Graph can be efficiently and successfully used on GPUs in such a way that applications are no longer limited to executing a single kernel at a time. The benefit has been proven by the analysis performed on the well-known Conjugate Gradient method, where we show the advantages of using CUDA and static graphs for this kind of problem. Additionally, by introducing simple code modifications (in contrast to other works implementing heavy optimizations) we are able to achieve an acceleration of up to $11\times$ w.r.t. the NVIDIA reference code thanks to a more efficient implementation, which reduces the overhead in the kernels launching and increments the potential overlapping of the kernels. We have also evaluated the use of static graphs and the OpenACC programming model, using the PSO algorithm as a test case. Several advantages arise from the use of OpenACC and CUDA Graph: (i) we provide

a mechanism to easily benefit from task-based programming using the GPU, without compromising performance; (ii) in most cases we are close to peak performance while comparing the results with a pure and optimized CUDA code; and (iii) by using OpenACC we allow programmers to write and offload parallel code into the GPU in an easy and transparent way. Multiple applications can benefit from using Static Graphs, not only in term of programability, but also providing a better performance. For instance, memory bound applications and/or iterative applications, such as those leveraged in this work, but also others, such as CFD simulations, image processing, AI or Linear Algebra kernels, are just some of the applications that can achieve a better performance by reducing the CPU–GPU communication overhead and achieving higher GPU occupancy. Finally, we propose a new pragma-based clause to integrate the use of static graphs as part of the OpenACC specification, which provides a simpler and more transparent way to implement static graphs. As shown in this paper, the potential in terms of coding productivity and performance offered by static graph offloading to accelerators is substantial and can be extended to directive-based programming models. Hence, we aim at influencing the adoption of this feature by further production applications and its incorporation in future releases of the OpenACC Specifications.

**Author Contributions:** L.T. contributed to conceptualization, methodology, software, validation, formal analysis, investigation, data curation, writing—original draft preparation and writing—review and editing. P.V.-L. contributed to conceptualization, methodology, validation, formal analysis, investigation, resources, data curation, writing—original draft preparation and writing—review and editing. J.S.V. contributed to investigation, writing—review and editing and supervision. A.J.P. contributed to conceptualization, methodology, investigation, resources, data curation, writing—review and editing, supervision, project administration and funding acquisition. All authors have read and agreed to the published version of the manuscript.

**Funding:** This research was funded by EPEEC project from the European Union's Horizon 2020 Research and Innovation program under grant agreement No. 801051. This manuscript has been authored by UT-Battelle, LLC, under contract DE-AC05-00OR22725 with the US Department of Energy (DOE). The US government retains and the publisher, by accepting the article for publication, acknowledges that the US government retains a nonexclusive, paid-up, irrevocable, worldwide license to publish or reproduce the published form of this manuscript, or allow others to do so, for US government purposes. DOE will provide public access to these results of federally sponsored research in accordance with the DOE Public Access Plan (http://energy.gov/downloads/doe-public-access-plan, accessed on 13 April 2022).

**Institutional Review Board Statement:** Not applicable.

**Informed Consent Statement:** Not applicable.

**Data Availability Statement:** Not applicable.

**Conflicts of Interest:** The authors declare no conflict of interest.

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
