# Peer review of "Towards Enhancing Coding Productivity for GPU Programming Using Static Graphs"

_electronics, doi:10.3390/electronics11091307_

Round 1

Reviewer 1 Report

In this paper, the authors proposed a method using the concept of Static Graphs in order to increase the coding productivity for GPU programming. To do so, the authors have combined the new CUDA Graph API with the CUDA programming model (including CUDA math libraries) and the OpenACC programming model. Authors use as test cases two different, well-known, and widely used problems in HPC and AI: the Conjugate Gradient method and the Particle Swarm Optimization. In the first test case (Conjugate Gradient) we focus on the integration of Static Graph with CUDA. In this case, we are able to outperform significantly the NVIDIA reference code, reaching an acceleration of up to 11× thanks to a better implementation, which can benefit from the new CUDA Graph capabilities. In the second test case (the Particle Swarm Optimization), we complement the OpenACC functionality with the use of CUDA Graph, achieving again, accelerations of more than one order of magnitude, and performance very close to a reference and optimized CUDA code. Our main target is to achieve a higher coding productivity model for GPU programming by using Static Graphs, which provides, in a very transparent way, better exploitation of the GPU capacity. The combination of using Static Graphs with two of the most important GPU programming models today (CUDA and OpenACC) is able to reduce considerably the execution time w.r.t. the use of CUDA and OpenACC only, achieving accelerations of up to more than one order of magnitude. Finally, we propose an interface to incorporate the concept of Static Graphs into the OpenACC specification. I read the article, it is a good article with right balance of theory and practice. This is well written and well-presented article; however, I do have the following concerns that need corrections during the revisions to improve this article further.

  • Please refine the title like, Towards Enhancing Coding Productivity for GPU Programming using Static Graphs.
  • Please write the contributions with bullets. Also, elaborate on the implications of this work.
  • Please provide more details of the code given in Listings 1, 2,3,4, and 5 with proper inputs and outputs.
  • Please provide a legend to clearly understand Figure 2. For example, what does red bar describe?
  • Please state the limitations of the proposed method.
  • Although the proposed method optimizes time, what about space complexity? The authors need to discuss this aspect in the revised work.
  • It would be better to include a table before the conclusion by including overall improvements of the proposed method compared to previous work.
  • It would be better to describe the main functions given in Figure 1 with a concise description in a table for clarity.
  • Please write notations in a table.
  • There exist a lot of inconsistencies regarding the usage of small and capital letters throughout the paper. For example, authors do not need to write the Graph with capital G.

Author Response

Dear Reviewer,

First, let me thank you the time and effort to review our paper. Your review has helped to improve the quality of our work.

Next, I summarized the modifications carried out on our paper according to your review:                                                                                                                                        

  1. We have refined the title.
  2. We used bullets to write contribution.
  3. We improved the pseudocodes by providing more details.
  4. We improved the legend of Figure 2.
  5. We clarified the contributions and the focus of our work.
  6. In this paper, we added new results, we didn’t improve the performance of previous results published in other publication, so no additional tables were necessary.
  7. We improve the explanation about the different functions involved in Listing 1.
  8. We reviewed the paper again, to remove any inconsistence found in previous versions

To make the review of the new content of the paper easier, all the modifications in the paper are in bold.

Thanks!
Pedro

Reviewer 2 Report

Below I am presenting my comments to the authors:

1/ The problem, the gap in the literature, must be better explored in the Abstract and Introduction. 

2/  Also in the Abstract, if possible, provide us numerical data. 

3/ The article is very very technical. Explore in a better way your scientific motivations in the Introduction. Also in the Introduction, present clearly your additions to the literature: what methods, not implementations, are you adding?

4/  What us the impact of the application type in the work? CPU-bound? IO-bound (mem, network, bus)?

5/  Is Section 5 the most important of the document? Present an architecture, design principles, Inout, processing and output diagram. 

6/  How about system scalability?

7/ Provide sa comparison table I the related work. 

8/ In the conclusion, present the contributions for the society: what a end-citizen could gain with your research?

9/ References are ok and adequate. 

Author Response

Dear Reviewer,

First, let me thank you the time and effort to review our paper. Your review has helped to improve the quality of our work.

Next, I summarized the modifications carried out on our paper according to your review:                                                                                                                                        

  1. We have improved abstract according to reviewer’s comments.
  2. We have clarified the methods and impact of our work in introduction and conclusion.
  3. We added a diagram to better explain the mapping or potential implementation of the new static graph OpenACC API on top of CUDA.

To make the review of the new content of the paper easier, all the modifications in the paper are in bold.

Thanks!
Pedro

Round 2

Reviewer 1 Report

The authors have addressed most comments in a satisfactory manner. The manuscript has significantly improved compared to the earlier version. Moreover, I suggest some minor corrections that will help strengthen the paper's quality.

  • Please write the conclusion section as one paragraph.
  • Please make the paragraph structure consistent. In the current form, some paragraphs are small and vice versa (especially, on pages 15 and 16).
  • It would be better to include a table before the related work section (i.e., at the end of Section #: 5) by including overall improvements of the proposed method with numbers and percentages.
  • Please change section 7 name from conclusions only to ‘Conclusion and Future Work’.
  • In conclusion, before introducing the future work, it would be better to write a few statements about the potential usage of the proposed implementation. For example, speeding up of large-scale analysis of social graphs, acceleration of cryptographic functions, and/or high-performance computing applications. Authors can possibly write some relevant application areas of the proposed implementation.

Author Response

We appreciate the time and effort carried out by the reviewer. The comments have helped to improve the quality and impact of our work.
We have carried out all the changes recommended by the reviewer
Please, find attached a new version of the paper, which includes all the modifications (in bold)

Reviewer 2 Report

Thank you authors for adopting all my previous recommendations. Now, we have a well-formated content to be published, with chances to be cited. In special, I would like to emphasize:

1/ In the Abstract, we have the problem and the contribution presneted in a clearer way.

2/ The Introduction was restructured to present the contributions of the work. Now, the reader can see the additions in the literature proposed by the article.

3/  The core of the work is presented in Sections 3 and 4. Here, we have graphs emphazing the results and contributions.

4/  Conclusion sections was reformulated to highlight again the contributions and how they were achieved. Future works were also updated in this part.

5/ References were updated too.

Considering this, at this moment I have the recommendation to approve the publication of the article.

Author Response

We appreciate the comments from the reviewer and the consideration that our work has the enough level to be published.

Please, find attached the last version of our paper which includes a few additions (in bold) that cover the comments of the other reviewer.
